# A Systematic Review on the Antimicrobial Properties of Mediterranean Wild Edible Plants: We Still Know Too Little about Them, but What We Do Know Makes Persistent Investigation Worthwhile

**DOI:** 10.3390/foods10092217

**Published:** 2021-09-18

**Authors:** Giulia Cappelli, Francesca Mariani

**Affiliations:** Institute for Biological Systems, National Research Council, Area RM1-(ISB-CNR), Strada Provinciale 35d, 9, Montelibretti, 00010 Rome, Italy; giulia.cappelli@cnr.it

**Keywords:** wild edible plants, antimicrobial effect, Mediterranean plant, Gram+ bacteria, Gram− bacteria, extraction protocols, bioactive compounds, essential oils

## Abstract

(1) Introduction: Bacterial resistance to antibiotics is estimated to be the cause of a major number of deaths by 2050 if we do not find strategies to slow down the rise of drug resistance. Reviews on Mediterranean wild edible plants (MWEPs) with antimicrobial properties are scarce in the main databases (PubMed, Scopus, and WoS). Hence, we proceeded to conduct a new review of the studies on MWEPs. (2) Methods: We used ‘wild edible plant’ and ‘antimicrobial’ as keywords. Within this group, exclusion criteria were reviews, studies concerning non-Mediterranean plants or non-edible plants, studies on topics other than plants or containing no description of antimicrobial properties, or off-topic studies. (3) Results: Finally, out of the one hundred and ninety-two studies we had started with, we reviewed thirty-eight (19.8%) studies concerning the antimicrobial properties of seventy-four MWEPs species belonging to twenty-five Families. Fifty-seven (77%) species out of seventy-four proved to be antimicrobial, with a stringent threshold selection. (4) Conclusions: Studies are still very heterogeneous. We still know too little about MWEPs’ properties; however, what we already know strongly recommends carrying on investigation.

## 1. Introduction

Bacterial resistance to antimicrobial drugs is an emerging threat [1]. Pathogenic and opportunistic bacteria in nosocomial-acquired infections more and more often cause complications in a postoperative period. This is even more worrying given the induction of immune suppression in modern medicine, the misuse of antibiotics in the latest fifty years, and the unbalance of healthy nutrients in the Western diet.

On the other hand, many people rediscover herbal medicine, which blurs the line between food and medicines—a line that, in many cultures, was never fixed and definitive.

In India, for example, the practice of modern medicine co-exists with indigenous traditional medicine, such as Ayurveda, Unani, and Siddha, which are extensively used by wide sections of the population [2]. Recently, Ayurveda’s immunity boosting measures were even recommended by the Indian government against SARS-CoV-2 infection. The principle behind such measures is that enhancing the body’s natural defense system (immunity) plays an important role in maintaining optimum health.

Additionally, Chinese herbal medicine (CHM) has a long history in the treatment of a variety of human diseases. Accumulating evidence shows that the clinical effects of CHM, too, are related to the up- or down-regulation of immune responses [3].

In this scenario, Mediterranean wild edible plants (MWEPs) and their antimicrobial properties have been known from ancient times, and nowadays, a growing number of people have rediscovered them as natural remedies for common infections [4].

In this respect, one of the problems concerning their use is the heterogeneity of the protocols used to extract and analyze the properties of their active components; unfortunately, such heterogeneity still marks the overall set of scientific studies on MWEPs, not to mention the enormous heterogeneity that characterizes the properties of plants at the outset (e.g., soil type, endophyte, sunlight UV, pathogen and pest attack) due to all the major influences upon secondary plant metabolism.

In this context, we reviewed the current literature on the medicinal value of Mediterranean native edible plants. Many of these are consumed because of the flavorings they provide (*n* = 20 in Appendix A).

### 1.1. Rationale

Before starting, in order to find reviews available in recent literature on wild edible plants with antimicrobial properties that grow in the Mediterranean basin, we searched through three databases: PubMed, Web of Science (WoS), and Scopus. By progressively filtering the number of matches with the first, second, and third keywords (i.e., ‘wild edible plant’, ‘antimicrobial’, and ‘Mediterranean’, respectively) we realized that, without adding any other time or geographic filters, the number of matches were just twenty (see Appendix A). Out of the final twenty retrieved reviews, five described only the ethnobotanical and/or nutraceutical properties of plants, while two described only the bioactive compounds of plants, two were book chapters, one was a conference paper, and one study was performed in Mexico and another in Denmark (neither of them being Mediterranean countries). Therefore, without any time range and geographic selection setting in the search engine, we could retrieve only eight reviews with our keywords.

In order to provide an updated review on MWEPs, we therefore decided to proceed with a new comprehensive analysis of the experimental studies describing this issue.

### 1.2. Objectives

The study’s objectives were as follows:

To provide an updated survey (considering the latest 20 years, see Appendix A) of those MWEPs that are still used in the daily diet by part of the Mediterranean population because of their antimicrobial properties;

To make a list of the MWEPs species proved to possess anti-bacterial, anti-fungal, and anti-viral properties;

To present the overall picture of the several protocols employed to extract the active components from the plants;

To try to establish whether a direct association between antioxidant and antimicrobial power was demonstrated in the reviewed studies;

To reinforce the belief that they are beneficial for human health and that they could be used more often in the daily diet;

To bring to attention of the scientific community the fact that, although we know too little about MWEPs properties, what we already know strongly recommends carrying on investigation.

## 2. Materials and Methods

### 2.1. Eligibility Criteria

The time interval we chose was 2001–2021—that is, twenty years during which the number of publications on WEPs and their antimicrobial properties grew significantly (two LOGs, from 10 to 1000 per year, see Appendix A). For only (Scopus) out of the three databases we examined was it possible to exclude India, China, USA, and Japan—four non-Mediterranean countries that produced a relevant number of publications on the topic. The research fields used for all the three databases were title, abstract, and keywords.

### 2.2. Information Sources

The three databases that were selected as sources from which to retrieve the scientific studies were PubMed (https://pubmed.ncbi.nlm.nih.gov, accessed on 14 April 2021), Web of Science (WoS) (https://apps.webofknowledge.com/WOS_GeneralSearch_input.do? accessed on 14 April 2021), and Scopus (https://www.scopus.com/search, accessed on 14 April 2021). We conducted the latest search on 14 April 2021.

### 2.3. Search Strategy

The first keyword, i.e., ‘wild edible plant’, allowed the retrieval of a very large number of studies (*n* = 4256), which we re-analyzed by using the second keyword (‘antimicrobial’, *n* = 292); subsequently, we noted the occurrence of duplicates and took removed them (*n* = 36). Finally, we re-analyzed the latter group of matches (*n* = 256) to detect the third keyword (Mediterranean, *n* = 192), as illustrated in Figure 1 and Appendix A.

### 2.4. Selection Process

Exclusion criteria for ineligible studies were review papers; papers that presented no plant, no edible plant, or non-Mediterranean plants; papers with no description of the plant’s antimicrobial properties; and papers that were otherwise off-topic (see Figure 1, PRISMA flow diagram and Appendix A).

Inclusion criteria for eligible studies were:Experiments performed in Mediterranean countries defined according to the biogeographical definition, which includes countries characterized by a Mediterranean climate and ecotype, even if they do not overlook the Mediterranean Sea (such as Portugal and Jordan);Experiments performed in non-Mediterranean countries but analyzing plants growing mainly in the Mediterranean basin (by checking the species geographical distribution on https://www.gbif.org/ accessed on 14 April 2021, see Appendix A). These plants, even if they are prevalently distributed in the Mediterranean basin, can also grow in other geographical areas, such as the case of *Sonchus* spp. in China, for example. This criterion means that we did not include experiments conducted with imported dried plants.

### 2.5. Data Collection Process

The significant narrowing of the number of studies included after the filtering process allowed each author to collect data from half of the thirty-eight reports, which were carefully distributed so as to ensure a balanced allocation of the same botanical families.

### 2.6. Data Items

We focused on antimicrobial properties data generated with at least one of the most well-constructed and widely employed assays (disk diffusion agar; minimal inhibitory concentration, or MIC; minimal bactericidal/fungicidal concentration, or MBC/MFC), which allowed us to include thirty-seven out of thirty-eight studies (see Table 1). The only study not employing one of those assays, either alone or in combination, but making use of a protocol similar to MIC (IC50, measuring the 50% growth inhibitory capacity) was reported as a single study.

### 2.7. Study Risk of Bias Assessment

We paid particular attention to the assays used to assess the inhibitory or microbicide properties of the MWEPs, as previously reported. The studies performing three antimicrobial assays (disk diffusion, MIC, and MBC) represented the most reliable group of experimental observations. However, in order to achieve the main goal of this review—that is, to give evidence of the antimicrobial power of MWEPs—we decided to group the results according to the analyzed bacterial or fungal species. In order to avoid unconscious bias when deciding how to include a study in this review, we relied on original experimental data that employed at least one of the assays listed above (for IC50, see the specification give above, in Section 2.6). In this way, the inclusion process would not be influenced by our judgement.

### 2.8. Effect Measures

We measured the results of the assays on antimicrobial properties by examining:The diameter of the growth inhibition zone on the agar plate in mm, for disk diffusion agar test. In accordance with Hudzicki [5], we adopted the following significance of the nearest whole inhibition zone, when obtained with an extract concentration ≤ 0.5 mg/mL.
**Diameter Zone, Nearest Whole mm****Resistant****Intermediate****Susceptible**≤1011–12≥13

Accordingly, we classified a microorganism with an inhibition zone ≥ 13 mm as one susceptible to a given extract.

2.The extract concentration of the minimal inhibitory concentration (MIC) and/or MBC/MFC in *w*/*v* (mg/mL) or in % (*v*/*v*). Minimum inhibitory concentration is defined as that which can block bacterial growth as long as the test compound is present in the growth medium. This means that the bacteria resume growth once placed on a medium without the test compound. When bacteria do not resume growth in the new medium, the compound displays a minimum bactericidal concentration instead.

To establish the value of MIC, the antibiotic is tested in a wide range of concentrations, from 0.002 to 256 μg/mL. For several antibiotics, the concentration of 0.5–1 μg/mL very often represents the threshold for classifying a bacterial pathogen as clinically resistant [6]. Because for medicinal plants the MIC is calculated at a 1000-fold more concentration (mg/mL), we chose to adopt a very stringent threshold, with a MIC ≤ 0.5 mg/mL, in order to make a rigorous selection of eligible studies. We therefore classified an extract with a MIC and MBC/MFC below that threshold as antimicrobial. For the volume/volume dilutions, we evaluated the experimental plan of each study.

3.The combination of 1 and 2, above.

Measurements ended up with a very stringent threshold, which we decided to adopt to classify the antimicrobial features of the MWEPs extracts. This means that across all of the studies, we classified the following as antimicrobial:In the studies with a disk diffusion test and the MIC, the extracts inducing a zone of inhibition ≥13 mm obtained with a MIC ≤ 0.5 mg/mL (in an assay made on a 100 mm plate, poured with 25 mL of Mueller–Hinton agar medium, to a measured depth of 4 mm; agar should be 1.7%, purchased from BD BBL, Franklin Lakes, NJ, as previously specified in the Kirby–Bauer disk diffusion susceptibility test protocol) [5];In the studies with MIC only, a MIC value ≤ 0.5 mg/mL;In the studies with disk diffusion test only, reporting the concentration of the extracts and an inhibition zone ≥ 13 mm, as obtained with a concentration value ≤ 0.5 mg/mL.

### 2.9. Synthesis Method

Having set the threshold of MIC concentration value and inhibition zone diameter to classify a MWEP as antimicrobial, as specified above in 2.8 (Effect Measures), we synthesized the study results according to the bacterial (Gram-negative and Gram-positive) and fungal species assayed with the extracts.

MWEPs species described in more than two studies were grouped together to compare their antimicrobial effects.

In the group of thirty-eight articles reviewed, the concentration of the efficacious extracts was not always clearly specified, thus creating a significant heterogeneity. Because our inclusion threshold was precisely based on MIC values (in mg/mL), we had to look for such data in the materials and methods section of each study very carefully. Several of them did not clearly specify the MIC concentration used to obtain a certain inhibition zone in the disk diffusion test, even if, in most cases, we could trace these data by analyzing the protocols adopted to prepare the extracts or to analyze their antioxidant properties.

### 2.10. Certainty Assessment

Over and above differences between studies, the comparison of the different MWEPs species against the same pathogen reassured us about the analyzed results. We obtained a reliable picture of data proving either (a) inhibiting or (b) not inhibiting effects by evaluating the number of studies assaying a single pathogen and being reasonably balanced between (a) and (b) type of results.

## 3. Results

### 3.1. Study Selection

In Figure 1, the PRISMA Flow Diagram describes the process of selection applied to the initial 292 studies retrieved by the respective search engines of the three databases consulted: PubMed (*n* = 61 matches), Web of Science (WoS) (*n* = 61 matches), and Scopus (*n* = 170 matches). Only thirty-six records were found to be duplicated in two or three databases, which brought the number of screened records to two hundred fifty-six (*n* = 256).

By analyzing the abstract of each retrieved manuscript, we realized that many of them did not satisfy the adopted inclusion criteria. Sixty-four (*n* = 64) studies were performed neither in Mediterranean countries nor with Mediterranean plants growing also in other geographical areas, although the three databases’ search engines retrieved them (final resulting set of 192 studies). Seventy-four (*n* = 74) studies did not describe the antimicrobial properties of the plants, but described instead their anti-inflammatory, anti-proliferative, or nutraceutical properties. Twenty-nine (*n* = 29) studies performed experiments with algae and mushrooms and not with plants. Thirty-seven (*n* = 37) were totally off-topic (for example, the wild edible plant was shown to be able to prevent adverse effects of a chemotherapeutic drug). One (*n* = 1) of the studies described a non-edible wild plant (fern), and eleven (*n* = 11) of them were reviews. As a result of this careful analysis, out of two hundred ninety-two studies retrieved, two hundred eighteen (*n* = 218) could not be included in further analysis; thus, we proceeded with the remaining thirty-eight studies (*n* = 38, 19.8% of 192 Mediterranean), in fact disclosing all the characteristics chosen at the beginning.

It is remarkable that the selection process, though performed with a rather stringent threshold for classifying antimicrobial activity, resulted in our review of less than 1% (0.89%) of the initial number of studies on WEPs (see the selection flow of articles in Appendix A).

### 3.2. Study Characteristics

In Appendix A, we show some examples of the geographical distribution of the WEPs species in the Mediterranean basin, starting with those existing mostly in the northern Mediterranean countries (Appendix A), followed by those diffused also in the Middle East countries (Appendix A) and those widely distributed also in the North African coast (Appendix A). To define a species as non-Mediterranean, we ascertained that there was no single accession in the entire Mediterranean basin (as in Appendix A for *Thymus daenesis*).

We grouped the MWEPs by their botanical families (Appendix A), and it is worth noting that eight families account for 69% of the species—namely, Asteraceae, Apiaceae, Brassiceae, Caryophyllaceae, Lamiaceae, Fabaceae, Polygonaceae, and Rutaceae.

The MWEPs species analyzed in more than one study are shown in Appendix A, being *Allium roseum* (Amaryllidaceae), *Centaurea raphanina*, and *Sonchus oleraceus* (Asteraceae), each of them described in four studies, and *Foeniculum vulgare* (Apiaceae), appearing in three studies; all together, they account for 39.5% of the experimental studies retrieved.

### 3.3. Risk of Bias in Studies

We paid much attention to the publication of authentic scientific information regarding plant extracts assayed on bacteria, fungi, and viruses. Surprisingly, we could not find a single study reporting anti-viral properties of MWEPs. Given the limited number of studies to be analyzed, we reported all the experimental data and protocols adopted to produce them. The significant heterogeneity of experimental procedures was considered and results were grouped according to the bacterial or fungal pathogen, which resulted as being sensitive to the plant’s microbicidal effect. A graphical summary describes the various starting materials (wild vs. cultivated, fresh vs. frozen-lyophilized or air-dried plants), the main solvents employed to prepare the extracts (alcoholic vs. aqueous extract, see Figure 2), and the antimicrobial assays performed (disk diffusion test vs. MIC and/or MBC, see Figure 3).

### 3.4. Results of Individual Studies

We report those few studies showing no antimicrobial effects at all (8), in which *Raphanus raphanistrum* did not meet our threshold criteria for being classified as antimicrobial (see Materials and Methods). In addition, *Eremurus spectabilis* extracts [22] did not pass the threshold of values established for classification of antimicrobial activity.

There was only one study exclusively using an IC50 assay in which eight MWEPs showed a very high antibacterial capacity, measured as less than 20 ppm [24].

There was only one in which the authors isolated the unique active compound cnicin, [20]—that is, in *Centaurea raphanina*, which was the only isolated compound displaying antifungal activity.

There were only two studies analyzing *Allium roseum* proteic extract [32,33], both of which did not display any antibacterial activity.

There was only one study undertaken with olive oils extracted from ripe and unripe fruit by mechanical pressing [34]; however, such oils did not show any antibacterial activity according to our criteria.

### 3.5. Results of Syntheses

We grouped the studies according to their antimicrobial effects.

#### 3.5.1. The Antimicrobial Effects Reported for the Most Studied Species

The sixteen MWEPs species analyzed in more than one study are listed in Appendix A. Fourteen out of sixteen (87.5%) displayed antimicrobial properties, as classified with our stringent threshold (see Materials and Methods).

Three species were analyzed in four studies: *Allium roseum*, *Centaurea raphanina*, and *Sonchus oleraceus.*

For *Allium roseum*, out of four studies, only one was undertaken with alcoholic and aqueous extracts [23], while the remaining three used protein extracts [27,32,33]. Interestingly, only the alcohol extracts (containing polyphenols) displayed a significant antimicrobial effect vs. many Gram-positive and Gram-negative bacteria, while the three protein extracts MIC were above the threshold we set up in this review.

For *Centaurea raphanina*, four studies compared the plant grown in situ, or wild, with the ex situ, or cultivated [17], through a further analysis of the soil enrichment effects on the plant antimicrobial properties [16,20,37]. The wild *C. raphanina* displayed the highest capacity to inhibit bacterial growth.

*Sonchus oleraceus*, with two studies on human pathogenic bacteria [7,8] and another two on pathogenic fungi [13,19], displayed relevant antimicrobial properties in all four independent studies.

*Foeniculum vulgare* was analyzed in three studies and displayed antibacterial IC50 values of less than 20 ppm [24] vs. *Staphylococcus mutans*, while it did not show relevant antimicrobial activities vs. other bacteria of fungi [29,31]. It is interesting that the only study showing antimicrobial properties was undertaken with plants collected from the mountains of the Nablus region and Kabul mountain (north Galilee), while the plants in the remaining studies were collected in the Sidi Bennour region (central Morocco, altitude 185 m above sea level) [29] and in sixteen locations in Tunisia [31], of which only two were at an altitude > 590 m above sea level.

In Appendix A, we show a Venn Diagram grouping the Gram-negative and Gram-positive common bacterial species assayed with this group of MWEPs species.

#### 3.5.2. The Overall Picture of MWEPs Antimicrobial Effects on Gram-Positive Bacteria, Gram-Negative Bacteria, Fungi

In Figure 4 we report the number of studies on Gram-negative (*n* = 27 species), Gram-positive (*n* = 18 species) bacteria and Fungi (*n* = 25 species) treated with MWEPs extracts.

How many Gram-negative bacteria are sensitive to MWEPs extracts?

Figure 4a shows all of the Gram-negative species and whether or not the MWEPs were efficacious. Overall, we found that 19 species out of 27 Gram-negative bacteria were sensitive to inhibition by the MWEPs extracts, and they represented 70% of the species analyzed in the thirty-eight studies of this review.

For the main Gram-negative bacteria of this group, which is composed of *E. coli*, *P. aeruginosa*, *S. typhimurium*, and *K. pneumoniae*, the studies with MIC ≤ 0.5 mg/mL were slightly fewer than those that were over this threshold. It is important to note, though, that these bacteria displayed a high incidence of nosocomial-associated antibiotic bacterial resistant (ABR) strains; as a matter of fact, *P. aeruginosa* and *K. pneumoniae* were included in the ESKAPE group (an acronym representing six nosocomial pathogens that exhibit multidrug resistance and virulence: *Enterococcus faecium*, *Staphylococcus aureus*, *Klebsiella pneumoniae*, *Acinetobacter baumannii*, *Pseudomonas aeruginosa*, and *Enterobacter* spp.) [45].

In this scenario the availability of several MWEPs species able to inhibit and eventually kill these dangerous bacterial species might represent a strategic reservoir of natural products for therapeutic interventions and disinfecting procedures (Appendix A). The reviewed studies did not show antimicrobial assays undertaken with clinically isolated and documented ABR strains. Nevertheless, the MWEPs’ property to inhibit the wild-type strains of nosocomial bacteria might contribute by lowering their replication and the growth of ABR strains. It goes without saying that studies undertaken with MWEPs on documented ABR nosocomial bacterial species are necessary.

It is also worth noting the very limited number of studies, i.e., one or four articles at most, on the Gram-negative species not sensitive to the MWEPs extracts. Nonetheless, this is a suggestion that future research be undertaken on these species, four of which are nosocomial-associated bacteria such as *E. cloacae*, *M. morganii*, *S. marcescens*, and *P. mirabilis*.

2.How many Gram-positive bacteria are sensitive to MWEPs extracts?

Overall, we found that 15 species, out of 18 of Gram-positive bacteria, were sensitive to the MWEPs extracts, representing 83% of the species analyzed in this review.

In Figure 4b, all of the Gram-positive species are shown and whether or not the MWEPs were efficacious. For the main bacteria of this group, which is composed of *S. aureus* Methicillin Sensible (MSSA), *B. cereus*, *E. faecalis*, *L. monocytogenes*, *B. subtilis*, and *S. epidermidis*, the studies with MIC ≤ 0.5 mg/mL were slightly more in number than those over this threshold. This is in accordance with the higher sensitivity of Gram-positive bacteria to antimicrobial drugs, but the use of natural products might also help to lower the antibiotic doses in human treatments.

The three species not inhibited by MWEPs extracts were analyzed only in one study each: they are environmental bacteria, such as *B. brevis* and *S. lutea*, or opportunistic bacterial infection in immune compromised hosts, such as *M. kristinae*. Again, the studies concerning these bacteria are very few.

3.How many fungi are susceptible to MWEPs extracts?

For fungal species, 88% were sensitive according to 38 studies.

In Figure 4c, all the fungi species are shown and whether or not the MWEPs were efficacious. It is quite clear that MWEPs were fairly efficient in inhibiting fungal growth, given that the number of studies demonstrating antimicrobial activity was higher than those proving the absence of such effect.

#### 3.5.3. The Comparison of MIC Values

Most studies analyzed both *E. coli* (Gram-negative) and *S. aureus* (Gram-positive); therefore, we could compare all the MIC values reported for these two main species (Figure 5).

It is well known that Gram-negative bacteria are more resistant to antibiotic treatments and that they embody 67% of the ESKAPE group of ABR bacterial species [45]. It is then important to note that the majority of MWEPs extracts displayed MIC values (Figure 5a) below the very stringent threshold adopted in our selection (see Materials and Methods).

Instead, *S. aureus* showed MIC values distributed at the turn of the threshold value (as shown in Figure 5b).

Only three studies analyzed the antimicrobial properties of the plants against clinically isolated strains [33,39,44]. It is worth noting that, in all the comparisons between collection species and the clinically isolated strains, the MIC values were not always higher for the latter (16 vs. 256 μg/mL for the most effective plants, such as *Chenopodium album*), but in some cases were significantly lower, even lower than conventional antibiotics (as for *Silene conoidea*, 0.01 vs. 0.1 mg/mL ampicillin).

Finally, there were only six studies using essential oils (EO), among which the EO distilled from *C. coronarium* [35] was the only one where MIC values were given in EO dilutions (*v*/*v*). Overall, the EOs were shown to be highly efficient in inhibiting bacterial and fungal growth [12,25,28,36], displaying in some cases, as compared with routinely employed antibiotics, a similar (vs. streptomycin) or even higher efficiency (vs. ampicillin) vs. Gram-negative bacteria [12]. The only exception was EO of *F. vulgare*, which did not show any antibacterial activity [31].

### 3.6. Reporting Biases

The thirty-eight studies reviewed were quite heterogeneous with respect to:-The protocols used to extract the active principles from the plants;-The assays employed to evaluate the antimicrobial properties of the extracts.

We therefore grouped the studies according to the bacteria against which they showed antibacterial properties in the presence of some MWEPs species.

We also sub-grouped the studies according to the assay reported (disk diffusion or MIC/MBC) to provide an overall picture of the most employed techniques (Figure 2 and Figure 3). This kind of approach could not bring to zero the bias in evaluating the study results, but it tried to present a wide picture of the range of antimicrobial properties of selected MWEPs against bacteria and fungi.

### 3.7. Antioxidant vs. Antimicrobial Properties: Direct or Inverse Association?

Even if most studies underpin the role of the antioxidant and reducing power of a MWEP by conferring it with effective antimicrobial capacity, there are still contradictory results to be evaluated. Overall, we cannot draw a conclusion yet.

For example, *Sonchus* species, with the highest reducing power, are the very same species displaying the highest antimicrobial properties [7]. *C. raphanina* cultivated plants have less polyphenol content than the wild plants, which correlates with their lower antimicrobial capacity [17].

In *Allium roseum* the authors found a direct association between antimicrobial properties and total phenolic compounds (TPCs) in the extracts made with different parts of the plant (either leaves, flowers, or bulbs) [21]. For *Silene* spp., the order of descending antimicrobial properties was compared with the respective order for metal chelating potential [39]. While it is true that the most antimicrobial species (*S. conoidea*) are also the most powerful in chelating metals, we must acknowledge that the descending order for both features in the six *Silene* species does not allow a direct association between them to be established. In another example [35], *P. hydropiper* proved to have a high level of antioxidant properties but weak antimicrobial properties: according to our threshold, none of the extracts were able to inhibit bacterial growth. In another study of MWEPs, the total antioxidant and free radical-scavenging activities of plant species showed a linear correlation with the total phenolics. For example, *M. polymorpha* was found to be the most active, and high antioxidant activity was observed for *G. laevigata*. Nonetheless, the antibacterial activity of methanolic extracts of all the plant species was lower than that of the positive control (streptomycin) against the tested bacterial species [41].

On the other hand, in a study on *C. macrocarpa*, it is interesting to note strong correlation values (0.7–0.9 and >0.9) between reducing power, total flavan-3-ols (TF3O), total phenolic compounds (TPC), and inhibiting properties vs. some Gram-positive bacteria tested (*E. faecalis*, *L. monocytogenes*, MRSA, and MSSA), even if the MIC values were over the threshold we set. Conversely, for DPPH scavenging activity, the strong correlation values were associated with total phenolic acids (TPA), total flavonols (TF), and inhibiting properties vs. *E. faecalis* and MRSA only [42].

Finally, the authors of another study wondered whether the results they had obtained could lead to the conclusion that there is no correlation between antibacterial activity against *S. mutans* and free radical scavenging [24]. However, an in-depth analysis revealed that the extracts of plants that exhibited an EC50 (amount of antioxidant necessary to decrease the initial DPPH absorbance by 50%) ≤ 100 ppm showed some degree of enrichment (4 orders of magnitude) in antibacterial activity.

From the analysis of the studies’ results, it is also evident that geographical location of plant collections (such as altitude in the case of *F. vulgare*) and extraction procedures also have substantial effects on the activity of the extracts [31].

### 3.8. Certainty of Evidence

Studies exhaustively documenting the antimicrobial properties against the very same bacterial and fungal species of pathogens were grouped and the data were summarized.

## 4. Discussion

### 4.1. MWEPs Open Questions

#### 4.1.1. After All, How Much Do We Know about Antimicrobial MWEPs?

It was rather surprising to find such a large number of studies erroneously included in the bulk of matches retrieved with the keywords ‘wild edible plant’ AND ‘antimicrobial’ by the three search engines (see Figure 1 and Appendix A). This must be borne in mind if we are convinced that MWEPs’ antimicrobial properties are widely studied just because we find a high number of articles using them as keywords.

In our research, out of one hundred and ninety-two initial matches, only 19.8% of them (thirty-eight) could be included in the detailed analysis of experiments describing the antimicrobial properties of Mediterranean WEPs.

Surprisingly, we could not find a single study reporting anti-viral properties of MWEPs, which poses the problem of increasing this kind of research activity.

Such a finding may be the result of the huge bias existing between public research studies, which are mostly focused on cultivated plant species, and the private studies concerning medicinal plants studied by pharmaceutical companies and never made public.

#### 4.1.2. How Many Are the Most Studied MWEPs?

Ten percent of all vascular plants are used as medicinal plants, and it is estimated there are between 350,000 and almost 500,000 species of them [46]. Other authors more conservatively estimated the existence of 1300 medicinal plant species in the EU, of which 90% are harvested from wild resources [47].

In this context, a review on seventy-four MWEPs, even if limited to the Mediterranean basin, immediately gives us an idea of how much the study of these plants is to be expanded.

#### 4.1.3. How Much of MWEPs Extracts Are Necessary vs. Antibiotics?

It is worth noting that for conventional antibiotics, in order to determine the value of MIC and MBC against different microorganisms, drugs are analyzed at a concentration of μg/mL (or mg/L). Conversely, in the case of MWEPs, the extracts for the antimicrobial assays are used at a 1000-fold-higher concentration, which is mg/mL (or g/L) [48].

Such a ratio might give us the erroneous impression that these plants are much less efficient than conventional antibiotics, but we must carefully consider some important differences between MWEPs extracts and drugs of synthesis.

The extracts reported in this review are documented to contain a heterogeneous mixture of compounds, ranging from phenolic, tannin, and flavonoids to minerals, fibers, organic acids, tocopherols, proteins, lipids, carbohydrates, free sugars, macro and micro elements (calcium, potassium, iron, manganese, etc.), and antinutrients (alkaloids, saponins, phytate).

The first group (collectively defined as polyphenols) is known to be the main set of compounds responsible for the antimicrobial properties of MWEPs; their concentration in the vegetal tissues is highly variable and constitutes a very small percentage of the total plant compounds (normally expressed in mg/g dry weight). This means that in order to reach a significant antimicrobial effect we must increase the extract concentration 1000-fold-times as compared with the antibiotic compound.

On the other hand, precisely because they are composed of many different active ingredients, these extracts do not allow bacteria or fungi to develop a resistance against their antimicrobial effects. On the contrary, the growing bacterial and fungal resistance against conventional antibiotic drugs, which are composed of a single molecule, has unfortunately been a public health emergency worldwide for many years now [1].

Nonetheless, in many studies MWEPs and antibiotics are used at the very same concentration (mg/mL) to compare their antimicrobial activities. In several cases, MWEPs extracts do work like conventional antibiotics (e.g., ampicillin), if not at lower concentrations [8,12,16,17,21,26,30,36,37,38,39,44], especially for bacterial species known for displaying ABR. In future studies, it will be advisable to analyze the effectiveness of the extracts formulated with different nanocarriers.

Additionally, it is necessary to state that future studies on MWEPs will also have to carefully examine the questions posed by various authors on natural products’ effective bioavailability and efficacy in vivo [49].

### 4.2. MWEPs Useful Properties

#### 4.2.1. Antimicrobial Efficacy

Gram-negative bacteria are sensitive to MWEPs extracts.

A total of 70% of the assayed Gram-negative species were inhibited by MWEPs extracts. For *E. coli*, most inhibiting MIC values were significantly below our stringent threshold. This effective antimicrobial property might suggest their possible use in co-administration with antibiotics as a means to fight Gram-negative bacterial infections.

Some of the Gram-negative bacteria not inhibited by MWEPs are opportunistic and responsible for urinary tract infections in immunocompromised hosts or in the event of catheter insertions, and the search for inhibiting natural products derived from MWEPs is definitely worthwhile.

2.Gram-positive bacteria are sensitive to MWEPs extracts.

A total of 83% of the assayed Gram-positive species were inhibited by MWEPs extracts. For *S. aureus*, the inhibiting MIC values were distributed at the turn of the threshold value, most of them being below the threshold.

3.Fungi are sensitive to MWEPs extracts.

Regarding fungi, 88% of the species analyzed in the thirty-eight studies were sensitive to MWEPs extracts. Approximately 300 fungal species on Earth are known to cause illnesses, such as *Candida* spp. and dermatophytes. In the food industry, bacteria and fungi cause problems during product processing and storage [50]. MWEPs could be a new safe and effective antimicrobial agent that could be applied in many fields.

#### 4.2.2. MWEPs vs. Antibiotic-Resistant Species

We report some examples of the advantages of the widening use of MWEPs extracts, also co-administered with conventional antibiotics to decrease the typically required drug amount and to contribute to slowing down the increase in antibiotic resistance [51].

For example, *Bacillus cereus* is an opportunistic bacterium that became a serious cause of nosocomial infections [4]. Nine studies (see Appendix A) reported in this review provide experimental data of MWEPs effective against this bacterium that could become pathogenic in immunocompromised hosts [8,11,12,16,17,30,32,33,39].

Additionally, regarding *Enterococcus faecalis*, which is responsible for urinary tract infections and under surveillance for resistance to aminoglycosides, six studies (see Appendix A) reported MWEPs extracts were effective in inhibiting its growth [9,21,33,36,39,44].

Another pathogen belonging to the ESKAPE group of ABR bacteria is *Klebsiella pneumoniae*, in which two studies (see Appendix A) reported effective inhibitory activity of MWEPs extracts [18,25].

Altogether, the studies here reviewed provide natural tools for inhibiting the growth of five out six ESKAPE pathogens, and the only one missing is an *Enterobacter*, for which we could include only one study.

#### 4.2.3. Antioxidant vs. Antimicrobial Properties

In the thirty-eight studies reported, there is still some controversy concerning the type of association between the antioxidant power of MWPs and their corresponding antimicrobial power, which is an issue deserving further investigation. Instead, what is clearer so far is the adjuvant role of natural antioxidant compounds in maintaining the good performance of the immune system [52,53]. This might suggest that even if the MWEPs antioxidant properties are not always the ones solely responsible for the killing of microbes, they certainly improve the capacity of the immune system to orchestrate the microbicide response of the infected host.

#### 4.2.4. Implications of the Results for Practice, Policy, and Future Research

MWEPs are being more and more rediscovered by consumers, both in Mediterranean and Northern European countries [4]. Their use in the daily diet might provide important micronutrients and healthy active components. We should start, then, to face the overwhelming heterogeneity that characterizes the way in which MWEPs are studied, in order to more systematically and consistently document their precious properties.

If it does not sound conceited, we would like to propose a list of steps to be shared and hopefully discussed inside the scientific community in order to reduce the great heterogeneity of the studies on MWEPs (as is also very often indicated in the EU Pharmacopoeia).

We should all put our best efforts forward to perform the very same tests to describe the antimicrobial properties of plant extracts. This means that at least the disk diffusion test and the MIC/MBC test should be included in our future studies, in order to allow comparison with similar studies.

Concerning the protocols for extraction, in order to allow comparison with similar studies, we should include at least alcoholic, hydro-alcoholic, and aqueous extracts from the very same part of the plant analyzed.

For aromatic plants, we should include the analysis of essential oils, which are highly effective in inhibiting bacteria and fungi.

Finally, concerning the pathogens analyzed, we should include at least the more characterized Gram-positive (*S. aureus*, MRSA) and Gram-negative (*E. coli*, *P. aeruginosa*) bacteria, which are responsible for the substantial number of infections in humans, in particular for community-associated infections.

#### 4.2.5. Limitations of the Evidence Included in the Review

Many studies using the disk diffusion agar test do not specify at which extract concentration they performed the experiments, which does not allow a comparison of the same assays in different studies with the same MWEPs species. The medium employed to perform the disk diffusion agar assay, which is also crucial for the results, is not always indicated.

Concerning fungi and certain bacteria, another issue is whether it is the vegetative cells or spores that are assayed. As it happens with bacteria, assays are often with planktonic cells rather than biofilms, which are very often responsible for antibiotic resistance in nosocomial-associated infections. Such an important issue must be addressed whenever possible.

## 5. Conclusions

We found that out of seventy-four MWEPs species, fifty-one (69%) belonged to eight out of the twenty-five botanical families analyzed in this review (see Appendix A). In particular, Asteraceae, Apiaceae, Brassicaceae, Caryophyllaceae, and Lamiaceae contained more than eight of the species most studied, and most of them displayed antimicrobial properties.

It is true that we still know very little about MWEPs, and it is necessary to standardize the protocols before further studying these plants.

It is advisable to avoid the extinction of MWEPs cultivation; however, to preserve the integrity of the active components, the agronomic practices must be as close as possible to the in situ growth.

The study of their effect on viruses must be increased.

On the other hand, MWEPs can inhibit both Gram-negative and Gram-positive bacteria and fungi. Importantly, the effective MIC on Gram-negative bacteria is significantly below the stringent threshold we employed, and some extracts do inhibit clinical isolates.

Their common antioxidant and metal chelating properties, despite some controversy, exert a positive effect on human and domestic animals’ immune systems, further helping them to face infections.

## Figures and Tables

**Figure 1 foods-10-02217-f001:**
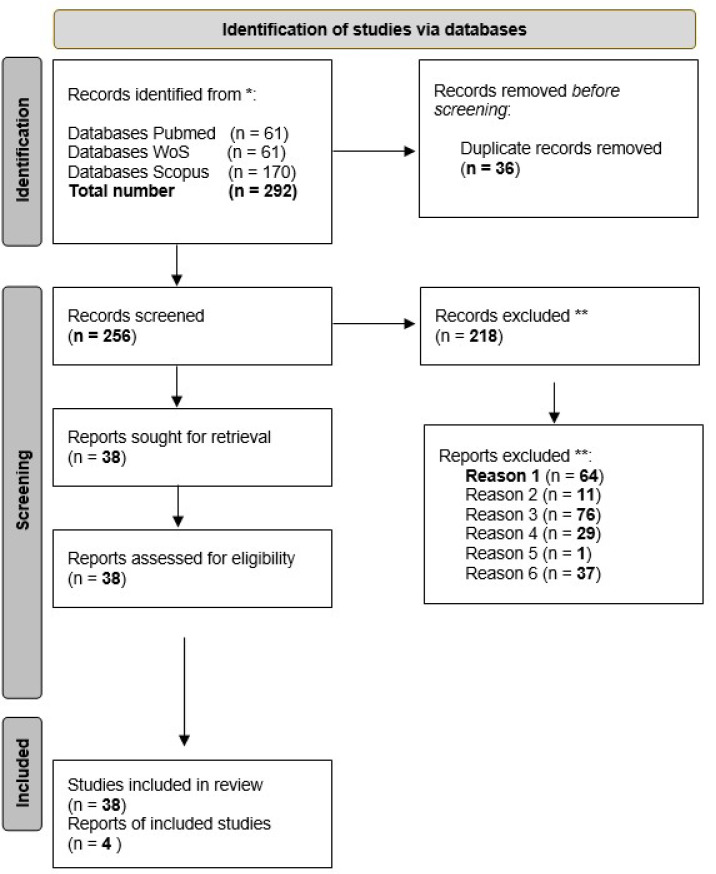
PRISMA 2020 flow diagram. ***** Number of records identified in each database. ** Exclusion criteria: Reason 1, non-Mediterranean plant (*n* = 64, ending up in a new initial number of 256 − 64 = 192); Reason 2, review (*n* = 11); Reason 3, properties other than antimicrobial ^§^ (*n* = 76); Reason 4, not plant (*n* = 29); Reason 5, not edible (*n* = 1); Reason 6, off-topic (*n* = 37). ^§^ Mainly studies describing the ethnobotanical use of the species against a wide range of non-transmissible diseases (diabetes, hypertension, chronic pain, tumors, etc.). Figure 2020. Statement: an updated guideline for reporting systematic reviews. BMJ 2021; 372: n71. doi:10.1136/bmj. n71.

**Figure 2 foods-10-02217-f002:**
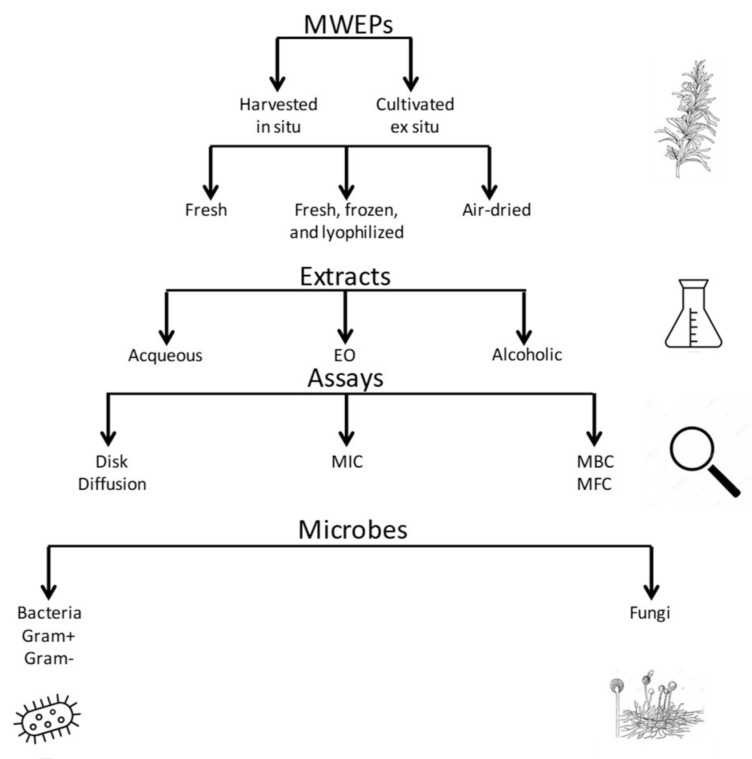
Graphical picture of the different types of plant materials, solvents for extracts, assays for antimicrobial properties, and pathogenic microbes (bacteria and fungi) tested.

**Figure 3 foods-10-02217-f003:**
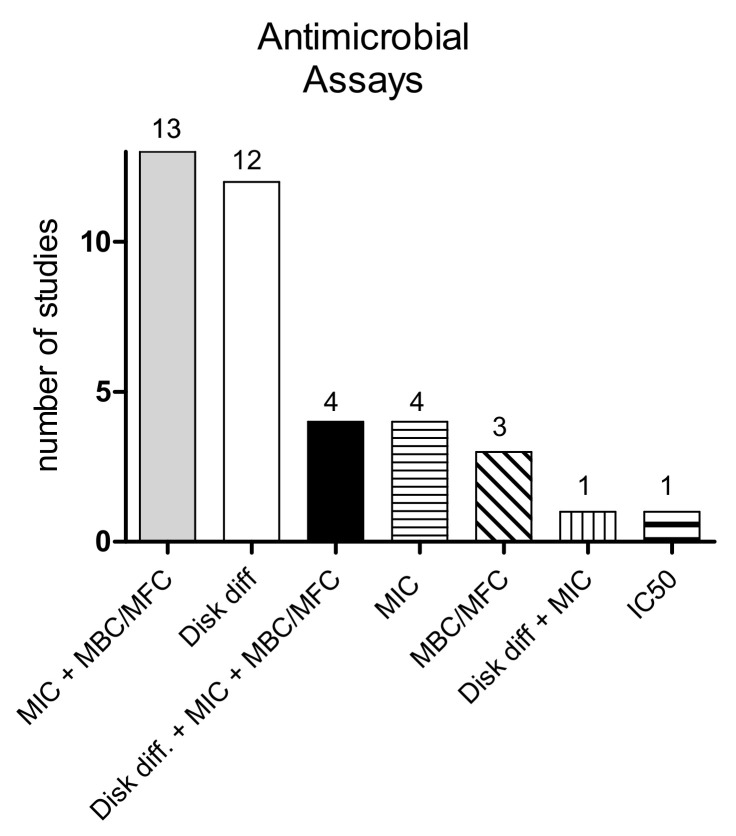
Graphical picture of the distribution of antimicrobial assays in the thirty-eight studies analyzed. A major portion of them (*n* = 36, constituting the 65.7%) employed either MIC, or MBC, or their combination also with disk diffusion agar, while 31.6% were based only on a disk diffusion agar test, and 2.6% (one single study) only on IC50 assay.

**Figure 4 foods-10-02217-f004:**
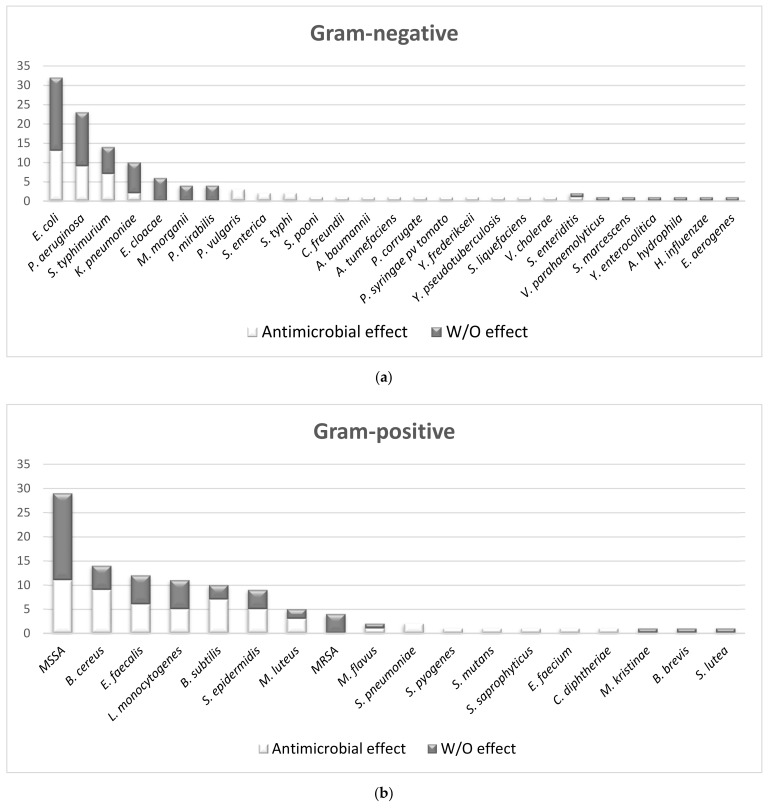
Number of studies (ordinate axis) of antimicrobial properties of MWEPs on Gram-negative (27 species), Gram-positive (18 species) bacteria and fungi (25 species). In different colors, we grouped the studies displaying an antibacterial effect (in white) and those in which no antibacterial (W/O, short for without) effect (in grey) could be included according to our thresholds. (**a**). Gram-positive bacteria; (**b**). Gram-negative bacteria; (**c**). fungi.

**Figure 5 foods-10-02217-f005:**
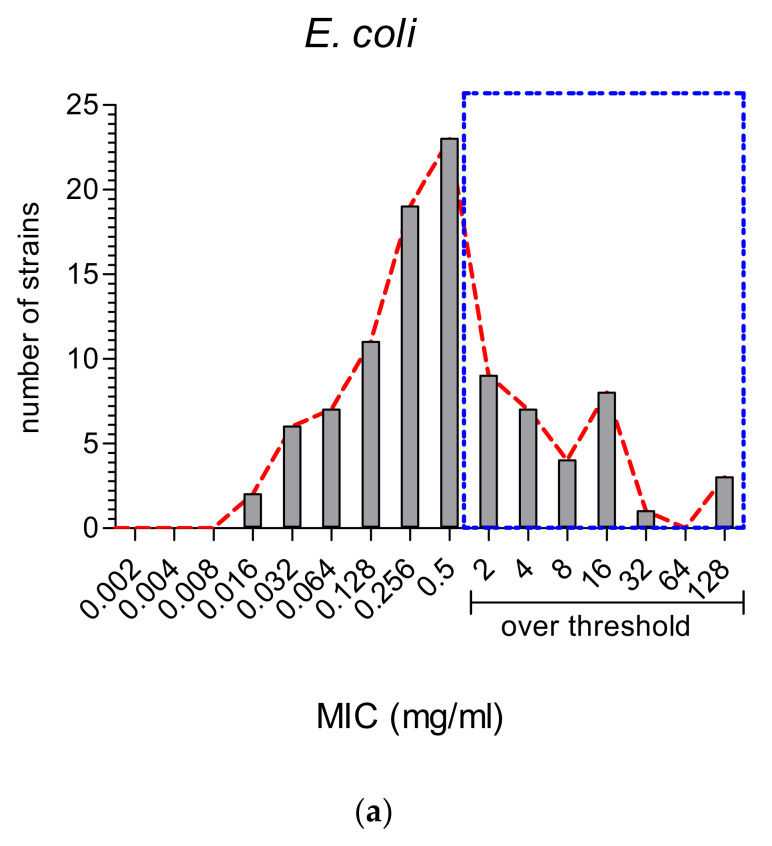
MIC values (in mg/mL) reported in the thirty-eight studies for MWEPs extracts vs. the two main and most studied pathogenic bacteria: (**a**) the Gram-negative *E. coli* and (**b**) the Gram-positive *S. aureus*.

**Table 1 foods-10-02217-t001:** List of the thirty-eight studies analyzed in this review, with a total number of 74 MWEPs species. In bold we indicate the species studied in more than one article. The species classified as antimicrobial by passing our stringent threshold were 57 (77%) in number and are marked with an asterisk (*) when associated with the reference study in which the antimicrobial properties were demonstrated.

Family	Species	Assayed vs.		Ref.
		Bacteria	Fungi	
Asteraceae	** *Sonchus oleraceus ** **	yes	no	[7]
	*Sonchus arvensis **	yes	no	“
	** *Sonchus asper ** **	yes	no	“
	*Sonchus uliginosus **	yes	no	“
Asteraceae	*Reicardia picroides **	yes	yes	[8]
	*Picris echioides **	yes	yes	“
	*Urospermum picroides*	yes	yes	“
	** *Taraxacum officinale ** **	yes	yes	“
	*Hymenonema graecum*	yes	yes	“
	** *Sonchus oleraceus ** **	yes	yes	“
	*Hedypnois cretica **	yes	yes	“
	***Taraxacum*** spp. *****	yes	yes	“
Fabaceae	*Ononis natrix **	yes	yes	[9]
Brassicaceae	** *Raphanus raphanistrum* **	yes	no	[10]
Asteraceae	*Bidens pilosa **	yes	no	[11]
Amaranthaceae	** *Chenopodium album ** **	yes	no	“
Apiaceae	*Heracleum pyrenaicum subsp. orsinii **	yes	yes	[12]
Asteraceae	** *Sonchus oleraceus ** **	no	yes	[13]
	** *Cichorium pumilum* **	no	yes	“
Portulacaceae	*Portulaca oleracea **	no	yes	“
Myrtaceae	*Psidium cattleianum*	yes	no	[14]
	*Psidium guajava*	yes	no	“
Apiaceae	*Scandix pecten-veneris **	yes	yes	[15]
Asteraceae	** *Centaurea raphanina ** **	yes	yes	[16]
Asteraceae	** *Centaurea raphanina* **	yes	yes	[17]
Asphodelaceae	** *Eremurus spectabilis ** **	yes	no	[18]
Boraginaceae	*Borago officinalis **	no	yes	[19]
Orobanchaceae	*Orobanche crenata **	no	yes	“
Plantagineceae	*Plantago coronopus **	no	yes	“
Plantagineceae	*Plantago lanceolate **	no	yes	“
Rosaceae	*Sanguisorba minor **	no	yes	“
Caryophyllaceae	** *Silene vulgaris* **	no	yes	“
Asteraceae	** *Sonchus asper ** **	no	yes	“
	** *Sonchus oleraceus ** **	no	yes	“
	** *Taraxacum officinale* **	no	yes	“
Asteraceae	** *Centaurea raphanina ** **	no	yes	[20]
Amaryllidaceae	** *Allium roseum ** **	yes	yes	[21]
Asphodelaceae	** *Eremurus spectabilis* **	yes	yes	[22]
Rutaceae	*Ruta angustifolia*	yes	yes	[23]
Apiaceae	** *Foeniculum vulgare ** **	yes	no	[24]
Lamiaceae	*Salvia palaestina fruticose **	yes	no	“
Lamiaceae	*Micromeria fruticose **	yes	no	“
Fabaceae	*Trigonella foenum-graecum **	yes	no	“
Asteraceae	***Cichorium pumilum*** jacq *	yes	no	“
Lamiaceae	*Salvia hierosolymitana* boiss *	yes	no	“
Rutaceae	*Ruta chalepensis **	yes	no	“
Asteraceae	** *Chrysanthemum coronarium ** **	yes	no	“
Lamiaceae	** *Ziziphora clinopodioides ** **	yes	no	[25]
Crassulaceae	*Umbilicus rupestris*	yes	no	[26]
Amaryllidaceae	** *Allium roseum ** **	yes	no	[27]
Lamiaceae	*Origanum syriacum **	yes	no	[28]
Euphorbiaceae	*Mercurialis annua*	yes	yes	[29]
Papaveraceae	** *Papaver rhoeas* **	yes	yes	“
Apiaceae	** *Foeniculum vulgare* **	yes	yes	“
Amaranthaceae	** *Chenopodium murale* **	yes	yes	“
Asteraceae	*Scolymus hispanicus*	yes	yes	“
Brassicaceae	*Sinapis arvensis **	yes	no	[30]
Polygonaceae	** *Polygonum aviculare ** **	yes	no	“
Asteraceae	** *Tragopogon aureus ** **	yes	no	“
Apiaceae	** *Foeniculum vulgare ** **	yes	yes	[31]
Amaryllidaceae	** *Allium roseum ** **	yes	yes	[32]
Amaryllidaceae	** *Allium roseum ** **	yes	yes	[33]
Oleaceae	*Olea europeae*	yes	no	[34]
Oleaceae	*Olea ferrugineae*	yes	no	“
Asteraceae	** *Chrysanthemum coronarium ** **	yes	yes	[35]
Amaryllidaceae	*Allium macrochaetum **	yes	yes	[36]
Asteraceae	** *Centaurea raphanina ** **	yes	yes	[37]
Polygonaceae	*Polygonum hydropiper*	yes	no	[38]
Caryophyllaceae	*Silene alba **	yes	yes	[39]
Caryophyllaceae	*Silene conoidea **	yes	yes	“
Caryophyllaceae	*Silene dichotoma **	yes	yes	“
Caryophyllaceae	*Silene italica **	yes	yes	“
Caryophyllaceae	*Silene supine **	yes	yes	“
Caryophyllaceae	** *Silene vulgaris ** **	yes	yes	“
Lamiaceae	** *Ziziphora clinopodioides ** **	yes	yes	[40]
Amaranthaceae	** *Chenopodium murale ** **	yes	no	[41]
Brassicaceae	*Eruca sativa **	yes	no	“
Brassicaceae	*Malcolmia africana **	yes	no	“
Malvaceae	*Malva neglecta **	yes	no	“
Fabaceae	*Medicago polymorpha **	yes	no	“
Fabaceae	*Melilotus officinalis **	yes	no	“
Brassicaceae	** *Nasturtium officinale ** **	yes	no	“
Apocynaceae	*Carissa macrocarpa*	yes	no	[42]
Apiaceae	*Smyrnium olusatrum*	yes	yes	[43]
Apiaceae	*Smyrnium perfoliatum*	yes	yes	“
Apiaceae	*Smyrnium rotundifolium* Miller	yes	yes	“
Apiaceae	*Smyrnium cordifolium* Boiss	yes	yes	“
Apiaceae	*Smyrnium connatum* Boiss and Kotschy	yes	yes	“
Apiaceae	*Smyrnium creticum* Miller *	yes	yes	“
Araceae	*Arum dioscoridis **	yes	yes	[44]
Amaranthaceae	** *Chenopodium album ** **	yes	yes	“
Malvaceae	*Malva sylvestris **	yes	yes	“
Lamiaceae	*Mentha longifolia **	yes	yes	“
Brassicaceae	** *Nasturtium officinale ** **	yes	yes	“
Papaveraceae	** *Papaver rhoeas ** **	yes	yes	“
Polygonaceae	** *Polygonum aviculare ** **	yes	yes	“
Polygonaceae	*Rumex acetosella **	yes	yes	“
Brassicaceae	*Sinapis alba **	yes	yes	“
Urticaceae	*Urtica dioica **	yes	yes	“

## Data Availability

Not applicable.

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
