# Peer review of "A Systematic Review on the Antimicrobial Properties of Mediterranean Wild Edible Plants: We Still Know Too Little about Them, but What We Do Know Makes Persistent Investigation Worthwhile"

_foods, 2021, doi:10.3390/foods10092217_

Round 1

Reviewer 1 Report

They did a good job on editing one or two suggestions still but this is one of the better jobs I have seen at doing the editing!
 there work plan is sound and they come up with useful hard facts and then suggestions 

have added some sticky notes of a comment nature  and they may want to add section or two to consider that 

I know there is a lot of general literature out there in this area  --i ran a whole 15 week course on this once   ie replacements for traditional antibiotics  

please find annotated manuscript   

Author Response

All the reviewer's requests were addressedmin the pdf file uploaded

Reviewer 2 Report

In general, the authors have addressed the main issues. In my opinion the manuscript can be accepted.

Author Response

The reviewer did not ask for other changes

This manuscript is a resubmission of an earlier submission. The following is a list of the peer review reports and author responses from that submission.

Round 1

Reviewer 1 Report

the authors have worked hard to summarize the results of extensive literature search for their topic

think there could be reduction in size of paper  eg section 4  repeats that already discussed   

liked the originality of section 3..7 

paper needs thorough editing 

Reviewer 2 Report

This manuscript describes a systematic review about the antimicrobial activity of wild edible plants. However, there are several issues that should be taken into consideration:

  • In the abstract 19,8% should be presented as 19.8%. This error was also present in various sections and tables of the manuscript. Please correct. The reference 1 should not be included in the abstract!
  • In section 1. introduction it was described that there are 20 reviews found. But this is very erroneous as these 20 reviews were found after the search of the articles and this information was given in the introduction. In addition, it was very disappointing that there are 20 reviews about this topic giving the idea that this topic was several times reviewed. But after some rapid analysis, it seems that these reviews were not describing the same information that was described in this manuscript.
  • In the section 2. materials and methods how do the authors determine the bias?
  • Relatively to the table 1 I suggest aggregate by each of the species (instead the aggrupation by reference).
  • In figure 4 which is W/O effect? Why some names of fungi are abbreviated and others such the Penicillium are not abbreviated? MSSA and MRSA abbreviations should be indicated in the footnote of the legend. The use of S.aureus MSSA was wrong because in the abbreviation was included the name of S. aureus. The figures should be improved as there are too many indications of the numbers and the lines to indicate the number of studies, in the title GRAM positive should be Gram-positive bacteria and the some for the GRAM negative should be Gram-negative bacteria. 
  • All the names of microorganisms should be changed e. g.  E.coli should be E. coli (with space). In addition, also should be reviewed the names of the bacterias and fungi  again as there are some errors (e. g. C. kruisei should be C. krusei)
  • Regarding the table 2 as suggestion it could be added but only as supplementary material as some of the information appeared before in the previous figures. In addition, the references should appear only at the column at the right. In this table, there are several errors (giving some examples Gram+ and Gram- in all the manuscript are appearing as Gram-positive and Gram-negative, sometimes ATCC appears within parentheses while in other cases not, while in others the ATCC was not indicated). Relatively to the species and bacteria some appears in bold while others not but was not indicated the reason. The indication of the tests used in the bacteria or fungi should appear on the right instead of the left  of the names. The indication of the extraction method/solvent/procedure also should be more concise. The use of vertical lines should be avoided.
  • In the section 3.5.1. The antimicrobial effects reported for the species most studies, relatively to the questions raised I think that should be removed. In addition, the third question was erroneously indicated as the number 2.
  • Relatively to the subsection "3.7. Antioxidant vs antimicrobial properties: direct or inverse association?" I think that was not described as an objective of this manuscript! 
  • In the subsection 4.2.4. Implications of the results for practice, policy and future research for several times the authors refer to the "editors" but I think that this could not be a task of the editors to request the authors to uniformize and use the same antimicrobial tests in the experimental studies for example. Thus, the authors should ameliorate the discussion.
  • In the section 5. conclusion I think that it could be more interesting to put 3 or 4 ideas in the conclusion remarks instead the use of the analysis SWOT.

Globally this manuscript needs careful reading and revision as several errors are present.